# Sex differences in sleep and influence of the menstrual cycle on women's sleep in junior endurance athletes

Maria Hrozanova[1]*, Christian A. Klöckner[2], Øyvind Sandbakk[1], Ståle Pallesen[3,4,5], Frode Moen[6]

1 Center for Elite Sports Research, Faculty of Medicine and Health Sciences, Department of Neuromedicine and Movement Science, Norwegian University of Science and Technology, Trondheim, Norway, 2 Faculty of Social and Educational Sciences, Department of Psychology, Norwegian University of Science and Technology, Trondheim, Norway, 3 Faculty of Psychology, Department of Psychosocial Science, University of Bergen, Bergen, Norway, 4 Norwegian Competence Center for Sleep Disorders, Haukeland University Hospital, Bergen, Norway, 5 Optentia, the Vaal Triangle Campus of the North-West University, Vanderbijlpark, South-Africa, 6 Faculty of Social and Educational Sciences, Department of Education and Lifelong Learning, Norwegian University of Science and Technology, Trondheim, Norway

* maria.hrozanova@ntnu.no

**Data Availability Statement:** All relevant data are within the paper and its Supporting information files.

## Abstract

Previous research shows that female athletes sleep better according to objective parameters but report worse subjective sleep quality than male athletes. However, existing sleep studies did not investigate variations in sleep and sleep stages over longer periods and have, so far, not elucidated the role of the menstrual cycle in female athletes' sleep. To address these methodological shortcomings, we investigated sex differences in sleep and sleep stages over 61 continuous days in 37 men and 19 women and examined the role of the menstrual cycle and its phases in 15 women. Sleep was measured by a non-contact radar, and menstrual bleeding was self-reported. Associations were investigated with multi-level modeling. Overall, women tended to report poorer subjective sleep quality ($p = .057$), but objective measurements showed that women obtained longer sleep duration ($p < .001$), more light ($p = .013$) and rapid eye movement sleep (REM; hours (h): $p < .001$, %: $p = .007$), shorter REM latency ($p < .001$), and higher sleep efficiency ($p = .003$) than men. $R^2$ values showed that sleep duration, REM and REM latency were especially affected by sex. Among women, we found longer time in bed ($p = .027$) and deep sleep (h: $p = .036$), and shorter light sleep (%: $p = .021$) during menstrual bleeding vs. non-bleeding days; less light sleep (h: $p = .040$), deep sleep (%: $p = .013$) and shorter REM latency ($p = .011$) during the menstrual than pre-menstrual phase; and lower sleep efficiency ($p = .042$) and more deep sleep (%: $p = .026$) during the follicular than luteal phase. These findings indicate that the menstrual cycle may impact the need for physiological recovery, as evidenced by the sleep stage variations. Altogether, the observed sex differences in subjective and objective sleep parameters may be related to the female athletes' menstrual cycle. The paper provides unique data of sex differences in sleep stages and novel insights into the role of the menstrual cycle in sleep among female athletes.

**Funding:** The authors received no specific funding for this work.

**Competing interests:** The authors have declared that no competing interests exist.

## Introduction

During sleep, the body and the brain undergo recuperative changes related to virtually all bodily systems required for athletic recovery [1, 2]. However, it is only when sleep is adequate in terms of its quality and quantity that recovery processes are optimal [cf. 3]. Previous research showed that increases in mental strain and training load can be disruptive to athletes' sleep, through associations with reduced sleep duration, rapid eye movement (REM) sleep and sleep efficiency [4–6]. However, athletes' sleep quantity and quality are also influenced by their sex [7–10].

Objective sleep data show that female athletes have more optimal sleep than male athletes, reflected in less time awake during the night, longer sleep duration, shorter sleep onset latency, and higher sleep efficiency in female compared to male athletes [8, 10, 11]. However, some gaps in our understanding of the sex differences in athletes' sleep remain, as the existing studies [i.e. 8, 11] spanned only 3–4 days, and did not provide insight into sleep stage distributions, which are important for optimal psychological and physiological functioning [12]. Moreover, a paradox exists in the sex-based sleep data. Despite the abundant evidence that women's sleep patterns are objectively better than men's, female athletes report a higher incidence of poor subjective sleep quality [13–15], and are 55% more likely to report sleep disturbances than male athletes [7]. Likewise, in two cross-sectional studies of athletes, female sex was found to be a significant predictor of poor subjective sleep quality [9, 13]. In this context, one frequently overlooked factor is the female menstrual cycle.

The female menstrual cycle occurs monthly throughout women's reproductive years. Each menstrual cycle lasts about 28 days, beginning with menstrual bleeding. The two main phases of the menstrual cycle include the pre-ovulatory follicular phase and the post-ovulatory luteal phase. During the follicular phase, the follicle-stimulating (FSH) and luteinizing hormone (LH) are released and stimulate the production of estrogens, eventually triggering a peak in LH. This allows for ovulation to occur at around day 14. Consequently, progestogens and estrogens are produced, peaking 5 to 7 days after ovulation. Given the absence of sperm implantation, their levels then decrease, resulting in menstrual bleeding, and the cycle continues [16]. Importantly, estrogen and progesterone receptors are found in the sleep-regulating areas of the brain [17, 18], which allows for a substantial hormonal influence on sleep in women.

After their first menstrual bleeding (menarche), girls are 2.75 times more likely to develop insomnia than boys [19, 20]. In a previous study on normally menstruating non-athlete women, objective sleep data were unchanged throughout the follicular and luteal phases—with the exception of REM sleep, which percentage increased in the follicular compared to late luteal phase [21]. Some women report pre-menstrual sleep disturbances [22], likely caused by pre-menstrual fluctuations in progestogens and estrogens [16]. Indeed, studies have found worsening of perceived sleep quality and sleep efficiency before or during menstrual bleeding [23–25]. However, evidence of unchanged subjective sleep quality across the menstrual phases also exists [26], thus the influence of the menstrual cycle on sleep remains unclear.

In the present study, we assessed subjective sleep quality, as well as objective sleep parameters, mental strain, training load and menstrual cycle in junior endurance athletes over 61 consecutive days. The primary aim was to investigate sex differences in objective sleep parameters, including sleep stage distributions, in junior endurance athletes, considering the possible confounding roles of mental strain, training load and subjective sleep quality. The secondary aim was to compare objectively measured sleep during menstrual bleeding and in the different phases of the menstrual cycle in naturally menstruating female endurance athletes.

## Materials and methods

### Participants

The present study was part of a larger research project investigating sleep in junior endurance athletes from high schools specialized for winter endurance sports in Norway [6]. In these schools, regular training is a part of athletes' educational plan, and they also often train after school with their respective sports teams. A total of 80 students were invited to information meetings about the research project, which took place in July 2018 at three schools situated in Trondheim, Meråker and Steinkjer (Mid-Norway), respectively. Athletes who were interested in participating in the research project were instructed to sign up for the study by contacting the researchers. Those interested then signed and returned a written informed consent form and were thus included. The data collection took part between August and September 2018. Due to equipment constraints (number of sleep monitors), the maximum number of participants that could be included was 60. Of these 60 initially included, four dropped out: of these, three were excluded due to lack of willingness to commit, while one participant did not provide a reason. Thus, 56 (93.3% of 60) athletes completed the study [6]. As the participants were self-selected, the sample should be regarded as a convenience sample. Although the representativeness of the sample as such is uncertain, the basic biological processes studied indicate that the results still are generalizable.

Of the 56 participating athletes, 37 were men and 19 women. For the investigation of sex differences, all 37 men and 19 women were included (mean age 17.6±.5 years). In order to elucidate the effect of menstrual cycle on sleep, men were excluded, as well as four women: two who used hormonal contraceptives, one who did not have menstrual bleeding during the study, and one who did not provide information on menstrual bleeding. Thus, for investigation of the effects of the menstrual cycle on sleep, data from 15 women were included (mean age 17.5±.5 years). None suffered from polycystic ovary syndrome, endometriosis, or severe premenstrual dysphoric disorder.

### Ethical considerations

All athletes provided written informed consent to participate. According to the Norwegian Health Research Act (§17.1), the lower legal age for providing consent to participate in health-related research is 16 years. Since all participants in this study were 16 years or older, parental consent was not necessary. The Regional Committee for Medical and Health Research Ethics (REC) in Central Norway approved the study (project ID 2017/2072/REK Central Norway).

### Instruments

**Somnofy sleep monitor.** Sleep was assessed with the Somnofy sleep monitor—a fully unobtrusive tool for sleep assessment based on impulse radio ultra-wideband (IR-UWB) pulse radar and Doppler technology. Validation against the gold standard of sleep measurement, polysomnography (PSG) was carried out in a healthy adult population and showed Somnofy to differentiate between sleep and wake and the different sleep stages with reasonable sensitivity and specificity. Epoch-by-epoch analyses showed that accuracy of measurements, determined by Cohen's kappa, was .97 for sleep, .72 for wake, .75 for light sleep (LS), .74 for deep sleep/slow wave sleep (SWS) and .78 for rapid eye movement (REM) sleep. For a full technical overview and results of the validation, including the limitations associated with the use of this tool, see Toften et al. [27]. In the current study, the following sleep variables were investigated: Time in bed (TIB), sleep onset latency (SOL), total sleep time (TST), time (hours, (h)) and %

**Table 1. Descriptions of the sleep variables collected with the Somnofy sleep monitor.**

| Sleep variable | Abbreviation | Unit | Description |
|---|---|---|---|
| Time in bed | TIB | h | Total time in bed, from arriving to bed before sleep onset to leaving in the morning after final awakening |
| Sleep onset latency | SOL | h | Time from lights off to sleep onset in any sleep stage |
| Total sleep time | TST | h | Total sleep time achieved during the night |
| Light sleep | LS | h / % | Time in the light stages of sleep (stage N1 and N2) / Light sleep standardized as % of TST |
| Deep/slow wave sleep | SWS | h / % | Time in the deep stages of sleep (stage N3) / Deep sleep standardized as % of TST |
| Rapid eye movement sleep | REM | h / % | Time in REM sleep (stage R) / REM sleep standardized as % of TST |
| Rapid eye movement sleep latency | REML | h | Time from sleep onset to the first REM stage |
| Sleep efficiency | SE | % | The ratio of time asleep (TST) to time between lights off and leaving bed |

For more information on sleep stages, see American Academy of Sleep Medicine [28].

of LS, SWS and REM, REM latency (REML), and sleep efficiency (SE). These sleep variables are described in greater detail in Table 1.

**Menstrual bleeding calendar.** Throughout the study, the female athletes kept a calendar overview of the 61 consecutive days comprising the data collection period. In the calendar, female athletes ticked off the days they experienced menstrual bleeding. Based on the self-reported menstrual bleeding days, the menstrual cycle was divided into 4 phases: Menstrual, pre-menstrual, follicular and luteal. Firstly, the menstrual phase was defined as the days of menstrual bleeding self-reported in the menstrual bleeding questionnaire. The pre-menstrual phase was defined as the 3 days prior to the first day of each menstrual bleeding [25]. Given the age of the participants, we assumed their menstrual cycles were ovulatory. All except two participants had two menstrual cycles during the study. Thus, the length of the menstrual cycle (day 1 of first menstrual bleeding to day 1 of second menstrual bleeding) was calculated individually for each participant. The length of the menstrual cycle was divided by 2 to identify the day of ovulation, which was used to differentiate between the luteal and follicular phases. Based on recommendations of Baker and Lee [16], the follicular phase was defined as the first half of the menstrual cycle, starting from the first day of bleeding. The luteal phase was defined as the latter half of the menstrual cycle, commencing after the last day of the follicular phase. In the two women that only had one menstrual bleeding during the study, a menstrual cycle length of 28 days was assumed, and the same principles as accounted for above were used for categorizing follicular and luteal phase. Thus, each cycle in each female participant was divided into menstrual (i.e. bleeding days) and pre-menstrual (i.e. the three days prior to onset of menstrual bleeding) phases, and follicular (i.e. the first half of the cycle) and luteal (i.e. the latter half of the cycle) phases. See S1 Fig for an illustrative example of the division of the menstrual cycle into phases.

In addition to reporting the days of their menstrual bleeding, the female athletes completed a short questionnaire about symptoms associated with menstrual bleeding. The first question asked about symptoms (changes in mood, stomach and back pain, headache, tender breasts, and other) generally experienced when menstruating. Participants answered "yes"/"no" for each symptom they experienced while menstruating. The second question asked whether, generally, they experienced that menstruation-related symptoms influenced their sleep ("yes"/"no"/"do not experience any menstruation-related symptoms").

**Well-being questionnaire.** A Well-being Questionnaire (WQ) was used to assess athletes' perceived mental strain. The WQ is a user-friendly, easy to administer self-report scale used for wellness and fatigue monitoring in athletes [29–32]. The original WQ consists of 5 questions asking about fatigue, sleep quality, muscle soreness, stress levels and mood (see S2 Table

for exact wording of the items and scale points). In the data collection of the present study, we were interested in obtaining a measure of mental strain with both affective and cognitive components [6]. Therefore, we amended the original WQ slightly, and changed the wording of the stress levels question into "worry / rumination". Athletes answered the WQ questions every evening before bedtime, on a visual analogue scale ranging from 0 (indicating low well-being) to 10 (indicating high well-being). Only the two questions about mental functioning were included in the analysis—one about mood (affective component of stress), and the other about worry/rumination (cognitive component of stress). These two questions were averaged to obtain a measure of mental strain. The WQ was built into the Somnofy app, which all participants had access to on their smartphones.

**Training diaries.** Participants recorded their daily training sessions (including type of training and training durations) in digital training diaries. This information was used to calculate training loads specific to endurance training as well as strength, plyometric and speed training, using the following methodology, applied also in Hrozanova et al. [6].

Athletes reported the time spent in endurance training intensity zones based on their own perception of training duration and intensity, in accordance with the five-zone intensity scale developed by the Norwegian Top Sport Centre (Olympiatoppen). Athletes' perception of duration and intensity is regularly entrained to physiological measures of heart rate and blood lactate during lab testing sessions. This method of self-reporting is common practice among winter endurance athletes in Norway, and it represents an accurate measure of training durations and intensities when validated against heart rate and blood lactate data [33].

The training intensity zones established by the abovementioned 5-zone intensity scale were then recalculated into a 3-zone scale, in order to match the zones with underlying physiological parameters [34]. The first and second lactate turning points were utilized for this purpose [34, 35], establishing three zones: (1) low intensity training (LIT), i.e. all training below the first lactate threshold, (2) moderate intensity training (MIT), i.e. training between the first and second lactate threshold, and (3) high intensity training (HIT), i.e. all training above the second lactate threshold [35]. Thus, from the original 5-zone scale, zones 1 and 2 were merged into LIT, zone 3 represented MIT, and zones 4 and 5 were merged into HIT.

Training impulse (TRIMP) was used to quantify training load based on durations in the reported training intensities (LIT, MIT, HIT) [36]. For endurance training, total duration in LIT was multiplied by 1, MIT by 2, and HIT by 3. For strength, polymetric and speed training, duration was multiplied by 1.5 –a constant based on the assessment of load by researchers and elite coaches in cross-country skiing. The respective TRIMP scores for each day were then summated, in order to calculate total TRIMP, which was utilized to quantify total training load. S1 Table shows the breakdown of the 5-zone intensity scale with typical heart rate and blood lactate values, the subsequent utilization of the 3-zone scale anchored in the first and second lactate turning point, TRIMP weights, reference information on how the zones correspond to different session ratings of perceived exertion [37] and examples of typical training sessions for each category [35].

**Pittsburgh Sleep Quality Index.** The Pittsburgh Sleep Quality Index [PSQI, 38] was used to measure subjective sleep quality. The index consists of 19 questions related to the experience of sleep quality during the past 30 days. In all, 15 questions were scored on a 4-point (0–3) Likert scale. The remaining 4 questions had open-ended response alternatives, which were scored on the same 4-point (0–3) Likert scale based on the given answers. A global composite score was calculated based on participant's answers ranging from 0 (indicating good sleep quality) to 21 (indicating poor sleep quality). The PSQI was adapted into Norwegian by Pallesen and colleagues (2005), and showed good psychometric properties [39]. Based on the current sample, Cronbach alpha was.69 (.72 for women, and.67 for men, respectively).

## Procedure

This study utilized a prospective observational design, in which sleep, mental strain, training load and menstrual bleeding were monitored daily over 61 consecutive days. Prior to study initiation, all participating athletes were invited to an information meeting about how to set up and use the sleep monitor. Information was provided about correct placement (up to 3 meters away from bed, and pointing at the participant's chest), connecting the device to wireless internet networks and Bluetooth, and information on how to troubleshoot technical issues. Athletes had one week to become familiar with the sleep monitor and the other instruments, but were also offered technical help throughout the study. The study commenced with completion of the PSQI. Each day throughout the data collection period, athletes used the sleep monitor, logged their levels of mental strain, signs of menstrual bleeding and training loads. In this way, we obtained insight into athletes' normal functioning, representative of the preparatory phase of the season.

## Statistical analyses

The collected data created a nested data structure, in which occasions of measurement were nested within participants, creating dependency among responses within each subject. To avoid excessive Type I errors and biased parameter estimates, statistical analyses were performed using multilevel modelling in Mplus, version 8.3 [40]. The use of Mplus allowed for the investigation of both within-level (repeated measurement units, level 1) and between-level (individual athletes, level 2) relations between study variables.

Random intercept models, which assume that the only variation between individuals is at their intercept, and that the effects of the explanatory variables were the same for each individual (fixed slope) were used. Intra-class correlation (ICC) coefficients, or the extent the dependent values of occasions of measurement in the same participant resemble each other as compared to those from different participants, were calculated. In multilevel models, ICC values quantify the relative magnitude of within and between-person variance. ICC values always fall between 0 and 1, with low values indicating a low proportion of between-level and high proportion of within-level variance whereas high values indicate a high proportion of between-level and low proportion of within-person variance, respectively [41, 42]. Associations at the within-level refer to the effects of the day-to-day variation within each athlete with the between-level effects removed. At the between-level, the results reflect the estimated variances of the predictor variables across athletes. For all multilevel models, $R^2$ values reflecting explained variance at the within-level were reported to estimate the magnitude of associations in the tested models. Guidelines for interpreting $R^2$ values were utilized to establish their practical relevance (scores above.04, or 4.0%, were deemed practically relevant) [43]. P-values were set at $< .05$ for all models.

Four sets of random intercept models were analyzed: (1) sex differences (binary predictor, 0 = male, 1 = female) in sleep (continuous outcome) in all participants; and in female athletes: (2) effects of menstrual bleeding vs. non-bleeding days (binary predictor, 0 = non-bleeding days, 1 = bleeding days) on sleep (continuous outcome); (3) menstrual vs. pre-menstrual phase (binary predictor, 0 = pre-menstrual phase, 1 = menstrual phase) effects on sleep (continuous outcome); and (4) follicular vs. luteal phase (binary predictor, 0 = luteal phase, 1 = follicular phase) effects on sleep (continuous outcome). Each set of random intercept models was applied to the sleep variables separately. In the first set of random intercept models, the effects of sex on mental strain, training load and PSQI were also investigated, in order to establish whether these variables should be included as covariates in the random intercept models.

## Results

### Sex differences in sleep patterns

Table 2 shows the associations between sex and objectively assessed sleep, mental strain, training load and subjective sleep quality. The results showed that women obtained longer sleep durations and higher SE than men. Sleep stages were also impacted, such that women obtained longer time in LS (h) and REM sleep (h), although SWS (h, %) remained unaffected. The proportion of REM, expressed as % of TST, increased concurrently. Further, the results showed that women obtained shorter REML than men. These results are graphically illustrated in Fig 1. All of the significant associations were deemed practically relevant, based on their respective $R^2$ values [43]. Sex explained 25% of the variance in TST, 8.4% of the variance in LS (h), 27.8% of the variance in REM sleep (h), 10.2% of the variance in REM sleep (%), 30.8% of the variance in REML, and 12.6% of the variance in SE, respectively. ICC values were based on a cluster size of 52.6 nights, and showed that 16–27% of the total variance in the sleep variables was due to differences between participants. Conversely, 84–73% of the total variance in the sleep variables was attributable to differences/variations within participants.

There were no sex differences in mental strain and total training load, thus these variables were not used as covariates in the subsequent regressions. Differences in subjective sleep quality trended towards significance ($p = .057$) and suggested that female athletes experienced worse subjective sleep quality than male athletes. Sex explained 6.1% of the variance in PSQI. For descriptive statistics on athletes' overall sleep patterns, mental strain and training load, see Hrozanova et al. [6].

**Table 2. Two-level random intercept regressions investigating sex differences (0 = men, 1 = women) in sleep and load variables, based on data from 37 male and 19 female endurance athletes.**

| DV | ICC | Est. men | S.E. men | Δ Est. women | S.E. women | *p*-value | $R^2$ |
|---|---|---|---|---|---|---|---|
| **Sleep variables** | | | | | | | |
| Time in bed (h) | .17 | 9.55 | .12 | .14 | .15 | .354 | 1.2% |
| Sleep onset latency (h) | .21 | .60 | .04 | -.06 | .07 | .433 | 1.2% |
| Total sleep time (h) | .17 | 7.37 | .08 | .51 | .11 | **< .001** | 25.0% |
| Light sleep (h) | .18 | 4.19 | .07 | .22 | .09 | **.013** | 8.4% |
| Light sleep (%) | .20 | 56.52 | .61 | -.75 | .85 | .377 | 1.2% |
| Deep/slow wave sleep (h) | .20 | 1.38 | .03 | .06 | .05 | .185 | 3.0% |
| Deep/slow wave sleep (%) | .18 | 21.51 | 1.07 | -.36 | .61 | .558 | .6% |
| REM sleep (h) | .16 | 1.81 | .03 | .23 | .05 | **< .001** | 27.8% |
| REM sleep (%) | .14 | 24.25 | .36 | 1.32 | .49 | **.007** | 10.2% |
| REM sleep latency (h) | .08 | 1.70 | .04 | -.27 | .06 | **< .001** | 30.8% |
| Sleep efficiency (%) | .27 | 77.58 | .89 | 3.92 | 1.30 | **.003** | 12.6% |
| **Load** | | | | | | | |
| Mental strain (au) | .52 | 3.63 | .18 | -.20 | .35 | .562 | .7% |
| Total training load (TRIMP) | .10 | 106.80 | 4.72 | -8.47 | 8.26 | .305 | 2.3% |
| **Subjective sleep quality** | | | | | | | |
| PSQI (au) | - | 3.11 | .34 | 1.10 | .58 | .057 | 6.1% |

REM = rapid eye movement; TRIMP = training impulse; PSQI = Pittsburgh Sleep Quality Index; DV = dependent variable; ICC = intra-class correlation; Est. = estimate; S.E. = standard error; $R^2$ = explained variance in %. Regressions were clustered on participant. As only one data point of PSQI was available, the analysis with PSQI was a linear regression, not multilevel. ICC is therefore not provided. Values are unstandardized. Significant results are highlighted in bold.

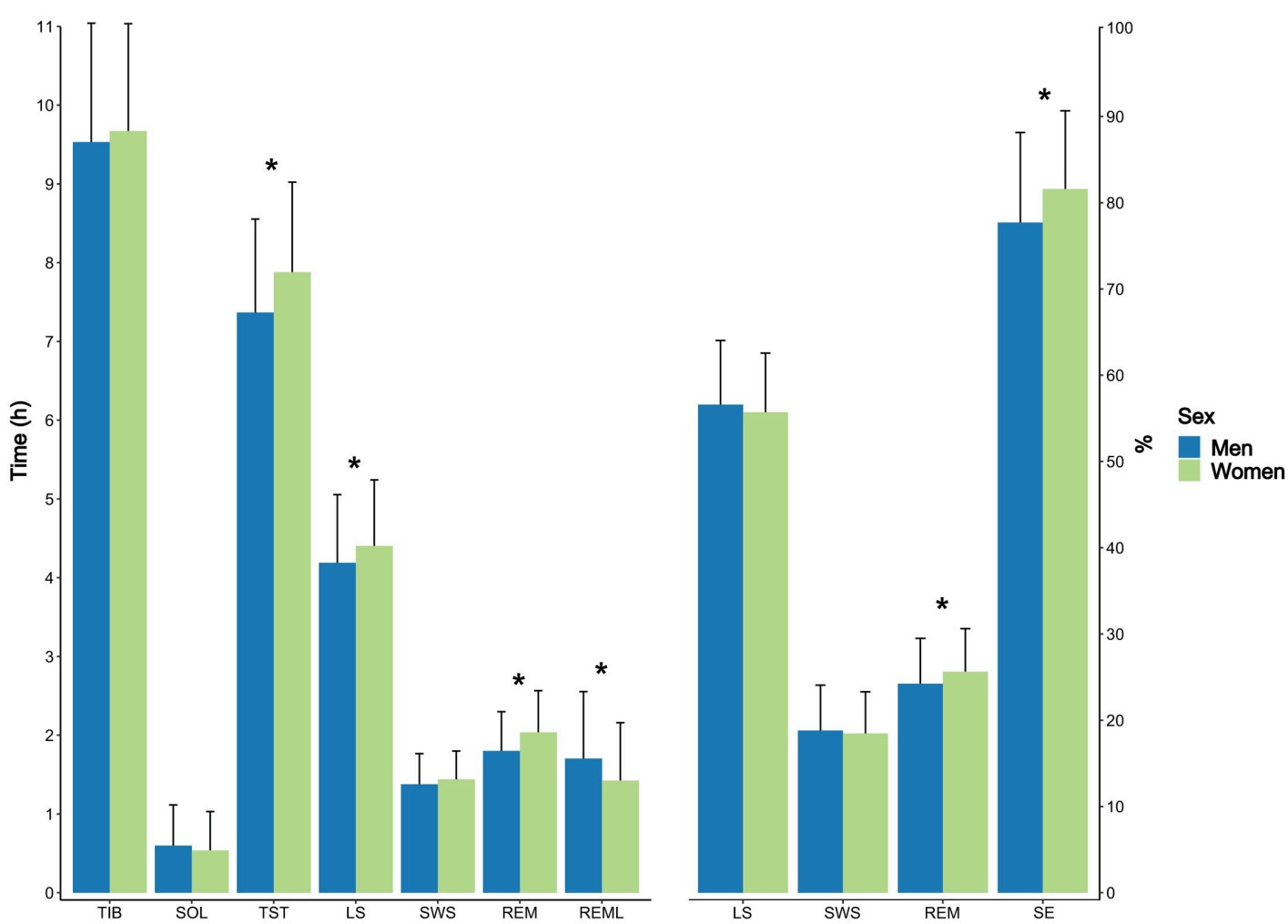

**Fig 1. Sex differences in objectively quantified sleep.** Abbreviations: TIB = time in bed, SOL = sleep onset latency, TST = total sleep time, LS = light sleep, SWS = deep / slow wave sleep, REM = rapid eye movement sleep, REML = REM latency, SE = sleep efficiency. Blue bars represent men, green bars women. Data is based on 37 male and 19 female junior endurance athletes. Whiskers represent standard deviations. * Represent significant differences between the sexes at p < .05.

### Menstrual bleeding-related sleep changes

The average length of the menstrual cycle in the 15 normally menstruating women was 28.3 ±4.5 days. The 15 women reported the following menstruation-related symptoms: 8 experienced mood changes, 12 experienced stomach and back pain, 3 experienced headache, and 4 experienced tender breasts. No other symptoms were reported. In all, 4 women reported that these symptoms influenced their sleep during menstruation.

Table 3 shows the within and between-level effects of menstrual bleeding days on sleep. At the within-level, menstrual bleeding days were associated with increases in TIB and SWS (h), and decreases in LS (%). These variations are graphically illustrated in Fig 2, red bars. On the between-level, there were significant variances in TIB, TST, LS (h and %), SWS (h and %), REM sleep (h and %), REML and SE. ICC values were based on an average cluster size of 54.3 nights per participant, and showed that 6–26% of the total variance in the sleep variables was due to differences between participants. Conversely, 94–74% of the total variance in the sleep variables was attributable to differences/variations within participants. The explained variances ($R^2$) were low (< .05-.7%).

**Table 3. Two-level random intercept regressions investigating the effect of menstrual bleeding (= 1) vs. non-bleeding days (= 0) on sleep variables, based on menstrual cycles recorded in 15 female endurance athletes.**

| DV | ICC | Within-level effect of menstrual bleeding days on the DV | | | | Between-level variance in the DV | | |
|---|---|---|---|---|---|---|---|---|
| | | Est. | S.E. | *p*-value | R² | Est. | S.E. | *p*-value |
| Time in bed (h) | .06 | .24 | .11 | **.027** | .4% | .12 | .04 | **.007** |
| Sleep onset latency (h) | .26 | .01 | .05 | .835 | < .05% | .06 | .04 | .144 |
| Total sleep time (h) | .07 | .04 | .06 | .469 | < .05% | .10 | .04 | **.016** |
| Light sleep (h) | .09 | -.11 | .07 | .121 | .2% | .07 | .02 | **.002** |
| Light sleep (%) | .13 | -1.62 | .70 | **.021** | .8% | 6.08 | 2.41 | **< .001** |
| Deep/slow wave sleep (h) | .19 | .07 | .03 | **.036** | .7% | .03 | .01 | **.012** |
| Deep/slow wave sleep (%) | .17 | .71 | .44 | .106 | .3% | 3.93 | 1.19 | **.001** |
| REM sleep (h) | .08 | .08 | .05 | .085 | .4% | .02 | .01 | **< .001** |
| REM sleep (%) | .07 | 1.00 | .54 | .065 | .6% | 1.70 | .68 | **.013** |
| REM sleep latency (h) | .05 | .06 | .08 | .496 | .1% | .03 | .01 | **.011** |
| Sleep efficiency (%) | .19 | -1.48 | .96 | .125 | .4% | 16.10 | 5.74 | **.005** |

REM = rapid eye movement; DV = dependent variable; ICC = intra-class correlation; Est. = estimate; S.E. = standard error; R² = explained variance in %. Regressions were clustered on participant. Values are unstandardized. Significant results are highlighted in bold.

## Menstrual phase effects on sleep

Table 4 shows the within and between-level effects of menstrual vs. pre-menstrual phase on sleep. At the within-level, menstrual phase was associated with decreased LS (h), % of SWS, and REML. These variations are graphically illustrated in Fig 2, yellow bars. At the between-level, there were significant between-persons variances in TIB, SWS (h and %) and SE. ICC values were based on an average cluster size of 54.3 nights per participant, and showed that 0–33% of the total variance in the sleep variables could be attributed to differences between participants. Conversely, 100–67% of the total variance in the sleep variables was attributable to differences/variations within participants. The explained variances (R²) were low (< .05–2.0%).

Table 5 shows the within and between-level effects of follicular vs. luteal menstrual phase on sleep. At the within-level, follicular phase was associated with increased SWS (%) and decreased SE. These variations are graphically illustrated in Fig 2, pink bars. At the between-level, there were significant between-persons variances in TIB, TST, LS (h and %), SWS (h), REM sleep (h and %), and SE. ICC values were based on an average cluster size of 13.5 nights per participant, and showed that 7–26% of the total variance in the sleep variables reflected differences between participants. Conversely, 93–74% of the total variance in the sleep variables was attributable to differences/variations within participants. The explained variances (R²) were low (< .05-.8%).

## Discussion

By investigating objectively monitored sleep over a period of 61 consecutive days, we aimed to identify sex differences in sleep among 37 male and 19 female junior endurance athletes. In addition, we investigated the associations between menstrual bleeding (based on self-reported bleeding days), phases of the menstrual cycle and sleep in 15 female endurance athletes. In the whole sample: (1) women obtained longer TST, shorter REML, spent longer time in LS (h) and REM (h, %), and had higher SE than men; within female athletes, (2) when compared to non-bleeding days, menstrual bleeding days were associated with increases in TIB and SWS (h), and decreases in LS (%); (3) when compared to the pre-menstrual phase, the menstrual

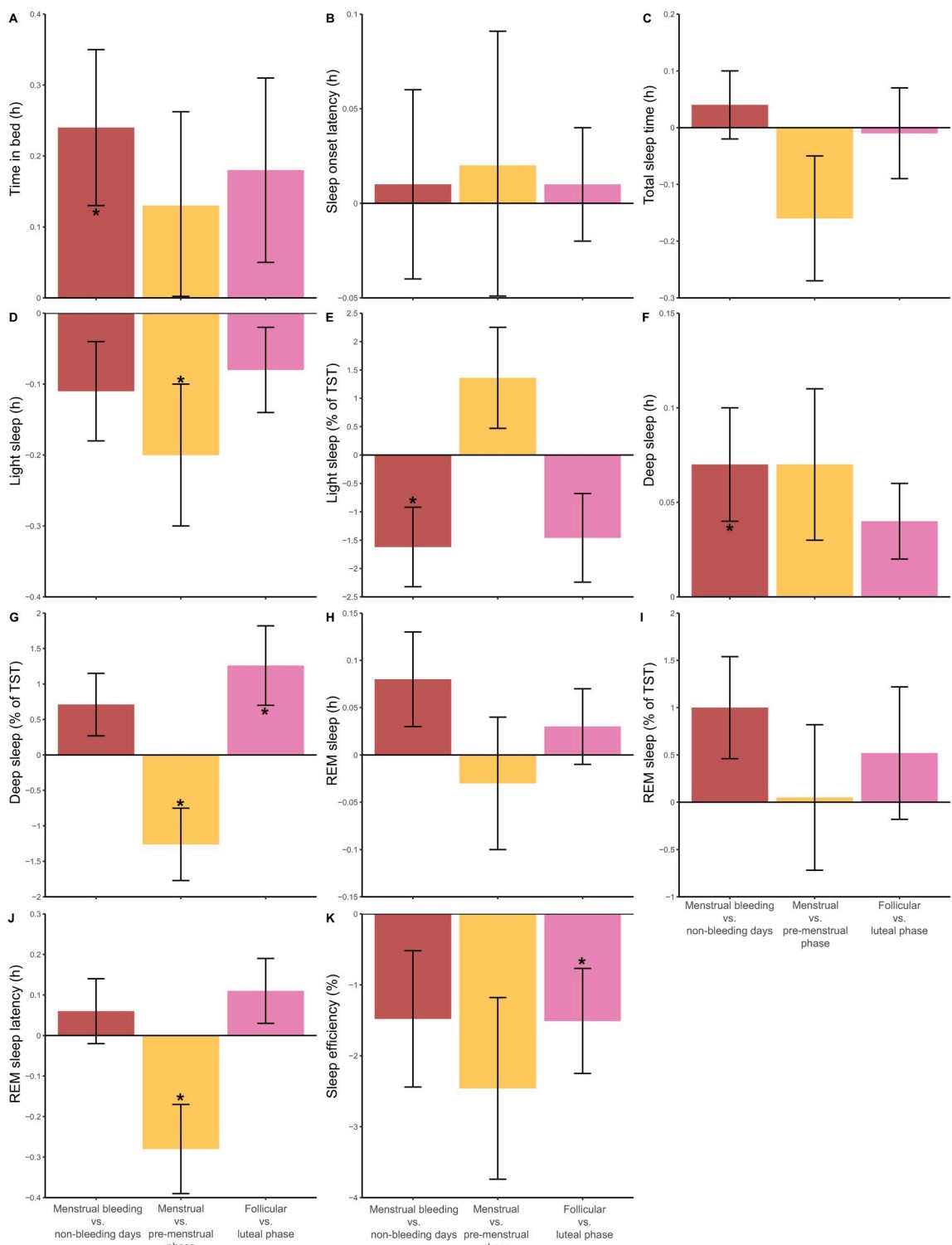

**Fig 2. Within-athlete variations in sleep variables across different phases of the menstrual cycle.** The measured variables included time in bed (A), sleep onset latency (B), total sleep time (C), light sleep in h (D) and % (E), deep sleep in h (F) and % (G), rapid eye movement (REM) sleep in h (H) and % (I), REM latency (J) and sleep efficiency (K), and spanned the following periods: menstrual bleeding vs. non-bleeding days (red bars), menstrual vs. pre-menstrual phase (yellow bars), and follicular vs. luteal phase (pink bars). Data are based on sleep monitoring in 15 female endurance athletes. The error bars represent the S.E.; * represents a significant change, p < .05.

**Table 4. Two-level random intercept regressions investigating the effect of menstrual (= 1) vs. pre-menstrual (= 0) phase on sleep variables, based on menstrual cycles recorded in 15 female endurance athletes.**

| DV | ICC | Within-level effect of the menstrual phase on the DV | | | | Between-level variance in the DV | | |
|---|---|---|---|---|---|---|---|---|
| | | Est. | S.E. | *p*-value | R² | Est. | S.E. | *p*-value |
| Time in bed (h) | .04 | .13 | .13 | .313 | .2% | .08 | .04 | **.024** |
| Sleep onset latency (h) | .33 | .02 | .07 | .825 | < .05% | .08 | .06 | .172 |
| Total sleep time (h) | .00 | -.16 | .11 | .146 | .5% | .00 | .05 | .954 |
| Light sleep (h) | .03 | -.20 | .10 | **.040** | 1.4% | .02 | .03 | .465 |
| Light sleep (%) | .11 | 1.36 | .89 | .126 | 1.1% | 4.49 | 2.91 | .122 |
| Deep sleep/slow wave (h) | .20 | .07 | .04 | .055 | 1.2% | .03 | .01 | **.024** |
| Deep sleep/slow wave (%) | .16 | -1.26 | .51 | **.013** | 1.8% | 4.04 | 1.93 | **.013** |
| REM sleep (h) | .04 | -.03 | .07 | .673 | .1% | .01 | .01 | .282 |
| REM sleep (%) | .08 | .05 | .77 | .953 | < .05% | 1.76 | 1.58 | .265 |
| REM sleep latency (h) | .04 | -.28 | .11 | **.011** | 3.3% | .02 | .02 | .359 |
| Sleep efficiency (%) | .22 | -2.46 | 1.28 | .054 | 2.0% | 19.85 | 7.54 | **.008** |

REM = rapid eye movement; DV = dependent variable; ICC = intra-class correlation; Est. = estimate; S.E. = standard error; R² = explained variance in %. Regressions were clustered on participant. Values are unstandardized. Significant results are highlighted in bold.

phase was associated with decreased LS (h), SWS (%) and REML; and (4) the follicular phase was associated with increased SWS (%) and decreased SE, when compared to the luteal phase.

Female athletes in the present study obtained longer TST and had higher SE than male athletes. The findings of longer TST and higher SE in women support previous evidence of sex differences in sleep among athletes [10, 11], and suggest that female athletes overall have better objective sleep indices than male athletes. Further, we confirm the existing paradox regarding objective vs. subjective sleep among female athletes established by previous research [13–15]: Female athletes tended to exhibit poorer self-reported sleep quality than male athletes. Although previous research found increases in mental strain and training load to be disruptive to athletes' sleep [4–6], we found no difference in mental strain and training load between sexes, suggesting that these variables do not explain the observed sex differences in sleep.

**Table 5. Two-level random intercept regressions investigating the effect of follicular (= 1) vs. luteal (= 0) phase on sleep variables, based on menstrual cycles recorded in 15 female endurance athletes.**

| DV | ICC | Within-level effect of follicular phase on the DV | | | | Between-level variance in the DV | | |
|---|---|---|---|---|---|---|---|---|
| | | Est. | S.E. | *p*-value | R² | Est. | S.E. | *p*-value |
| Time in bed (h) | .07 | .18 | .13 | .184 | .5% | .12 | .04 | **.006** |
| Sleep onset latency (h) | .26 | .01 | .03 | .724 | < .05% | .06 | .04 | .148 |
| Total sleep time (h) | .07 | -.01 | .08 | .945 | < .05% | .10 | .04 | **.017** |
| Light sleep (h) | .10 | -.08 | .06 | .207 | .2% | .07 | .02 | **.002** |
| Light sleep (%) | .10 | -1.46 | .78 | .061 | 1.3% | 4.72 | 2.23 | **.034** |
| Deep/slow wave sleep (h) | .19 | .04 | .02 | .155 | .3% | .02 | .01 | **.014** |
| Deep sleep/slow wave (%) | .11 | 1.26 | .56 | **.026** | 1.7% | 2.92 | 1.64 | .076 |
| REM sleep (h) | .08 | .03 | .04 | .383 | .1% | .02 | .01 | < **.001** |
| REM sleep (%) | .01 | .52 | .70 | .457 | .3% | 2.36 | .88 | **.007** |
| REM sleep latency (h) | .02 | .11 | .08 | .203 | .5% | .01 | .01 | .140 |
| Sleep efficiency (%) | .20 | -1.51 | .74 | **.042** | .8% | 16.61 | 6.04 | **.006** |

REM = rapid eye movement; DV = dependent variable; ICC = intra-class correlation. Regressions were clustered on participant. Values are unstandardized. Significant results are highlighted in bold.

Taken together, we show that female athletes have better objective sleep but tend to experience worse subjective sleep quality than male athletes, which cannot be attributed to differences in mental strain and training load.

The present study is the first to identify sex differences in sleep stages among athletes. Female athletes obtained shorter REML, spent longer time in LS (h) and REM (h, %). Comparing the present results with sleep data from the general population, we attest to the finding of longer REM (h and %) among women. However, in the general population, men typically obtain lighter sleep than women, exemplified by more LS and less SWS [44]. This stands in contrast with the present results, which showed that female athletes obtain more LS (h) than male athletes. Interestingly, SWS (h and %) was not different between the sexes, possibly reflecting the importance of SWS for athletic recovery, irrespective of sex [45]. The finding showing shorter REML among female athletes is novel and previously unexplored in the context of athletics. Variations in REML are typically discussed in clinical populations (i.e. depression [46], excessive sleepiness [47]), which is not representative of the current population of junior athletes. Among healthy sleepers, studies either found no sex differences in REML [48], or shorter REML among women than men [49]. However, the mechanisms behind these differences are unclear. In the present study, sex alone explained a moderate amount of variance in TST, REM (h) and REML, reaching 25.0%, 27.8% and 30.8%, respectively. Based on existing guidelines for interpreting effect sizes, these effects are of moderate practical significance [43]. Thus, TST, REM (h) and REML account for the largest sex differences in female and male athletes' sleep.

One possible reason for the paradox in women's objective versus subjective sleep parameters may be the hormonal fluctuations and negative side-effects associated with the menstrual cycle. By comparing days with menstrual bleeding to non-bleeding days, we found that menstrual bleeding days were associated with increases in TIB and SWS (h) and decreases in LS (%). In a population of healthy, young, normally menstruating women, Driver et al. [21] found no such associations. It is therefore possible that the present results may be unique to the athletic population. Female athletes may require more "downtime" in bed during menstrual bleeding as evidenced by the increased TIB, perhaps because of more discomfort or lethargy due to blood loss. Interestingly, the increase in SWS (h) during menstrual bleeding was accompanied by a decrease in LS (%). Therefore, it may be that the variations in SWS (h) represent a compensatory mechanism, whereby the time spent in SWS (h) increases, and the proportion of LS (%) decreases. The observed variation in LS was not apparent in when expressed in terms of duration (h), only in regard to proportion of TST (%). As SWS is especially important for physiological recovery [50], we hypothesize that menstrual bleeding may pose an additional load on the female body, inciting a greater need for physiological recovery, which may occur at the expense of LS. This may occur, at least partly, due to the variety of menstruation-related symptoms (mood changes and pain) that most of the participating female athletes reported. Indeed, 4 of the 15 participating female athletes reported that these symptoms influenced their sleep during menstruation, which may have influenced athletes' objective sleep indices. Monitoring athletes' discomfort and the severity of negative side-effects experienced during menstruation should be emphasized in future research. In addition, future research should monitor the subjective need for sleep throughout the menstrual cycle, including biological (e.g. muscle damage and inflammation biomarkers [51]) and mental factors (e.g. depressive or pre-menstrual symptoms [52]) that may play a role in the associations between menstrual bleeding and sleep.

Investigating the effects of menstrual phases on sleep in female athletes, we found a decrease in LS (h), SWS (%) and REML, in menstrual when compared to pre-menstrual phases. These are the first results among female athletes to provide insight into variations in sleep across

menstrual cycle phases, indicating that sleep/wake parameters (i.e. TIB, TST, SOL, SE) remain stable, while aspects of sleep architecture (i.e. sleep stages) seem to be affected. During the menstrual phase, changes in female athletes' sleep architecture indicate overall lighter sleep (less LS in minutes but preserved % in relation to TST, decreased proportion of SWS in relation to TST, unchanged REM) and earlier REM onsets. However, the low $R^2$ values indicate that these findings are of little clinical relevance, which is in line with conclusions of previous research [24]. The present findings are in contrast to one study utilizing PSG to compare sleep stage distribution across different menstrual phases in a young non-athletic population [21]. Other studies also provide conflicting results, such as poorer subjective sleep quality during pre-menstrual phase, which also coincided with longer self-reported TSTs [23], and poorer subjective sleep quality both pre-menstrually and during menstruation [25]. However, the findings of the present study may not be directly comparable with previous research, due to methodological (i.e. assessment of sleep and the menstrual cycle) and sample differences. In order to increase our knowledge about the influence of the menstrual cycle on sleep in female athletes, we are dependent on more research conducted in athletic samples, including standardized test protocols, as well as objective measures of sleep and menstrual phases Finally, SE decreased and SWS (%) increased during the follicular phase, when compared to the luteal phase. The follicular phase represents the first half of the menstrual cycle, which also encompasses the days of menstrual bleeding. In the present study, variations in SWS are apparent during the days of menstrual bleeding and in the menstrual phase, indicating changes in physiological recovery among junior female athletes during periods associated with menstrual bleeding. Interestingly, mean SE among the women participating in the Driver and colleagues' study [21] was somewhat higher than in the present sample (91.6% vs. 81.5%). The correspondence in findings occur despite the fact that Driver and colleagues [21] utilized obtrusive PSG equipment, which may have contributed to the decreased SE during the follicular phase. Alternatively, the previously discussed symptoms related to menstrual bleeding, or overall decreased physical comfort, may possibly lead to more fragmented sleep and thus lower SE. One previous study showed elevation of muscle damage and pro-inflammation biomarkers during extended recovery from intensive endurance exercise performed in mid-follicular phase. These biomarker findings were hypothesized to indicate a greater need for recovery during the mid-follicular phase [51], which is in line with the present findings. However, our findings contradict previous research which showed a decrease in SE [53], REM [21] and REML [54] during the luteal, as compared to follicular phase. One other study found variations in SWS and awakenings throughout the menstrual cycle, but post-hoc tests failed to support the sleep-related differences between phases [55]. These inconsistencies may be due to methodological differences (e.g. investigating the associations in non-athlete females [21, 53, 54]; use of subjective sleep diaries [53], and more accurate methods of assessing menstrual cycle phase [21, 54] or sleep stages [21, 54]) or variations in sample sizes. It has previously been postulated that in normally menstruating women, it is typical to observe "little or no consistent changes in sleep between follicular and luteal phase" [21]. The low explained variance shown in the present study also points to a limited predictive power for detecting sleep differences between the follicular and luteal phases. Based on existing guidelines, these results are below the cutoff for practically significant effects [43].

## Limitations

The present study had some limitations which should be kept in mind when interpreting the results. Firstly, menstrual bleeding was self-reported. Thus, ovulation and timing of the menstrual phases in each female athlete were inferred from the self-reported menstrual

bleeding days and not by objective assessment. Ideally, self-reported menstrual bleeding days should be combined with objective methods to ascertain ovulation (assessing hormone levels from urine), and to determine menstrual cycle phase (venous blood samples to determine serum hormone levels) objectively [56]. This may be especially important in athletic populations, as 30%-50% of physically active women have been shown to have anovulatory, or luteal-phase deficient menstrual cycles [57, 58]. Secondly, the radar device used in the present study to monitor sleep has been validated against PSG on both sleep/wake and sleep stage parameters and showed high accuracy in healthy young adults. Despite high accuracy, there was some divergence in agreement, with an average Cohen's kappa of .63 on the tested parameters [27], and a degree of measurement error should therefore be taken into consideration when interpreting the present results. Lastly, low proportions of explained variances in the analysis of menstrual phase effects on sleep were observed, possibly reflecting the sample of healthy young athletes not suffering from polycystic ovary syndrome, endometriosis, or severe premenstrual dysphoric disorder [16]. Moreover, the questions about menstrual symptoms among the female athletes were asked in general, and not in association with the specific menstrual bleedings that occurred during the data collection period. A detailed account of how menstrual symptoms affect functioning during the daytime, including intake of pain medication for symptom alleviation, should be investigated in future research. It is also likely that several other variables, not controlled for in this study, may have influenced the associations at play. Such variables may include daily measures of subjective sleep quality, the regularity of sleep/wake patterns, chronotype, or openness about menstrual bleeding and its effects on training in the team with coaches and teammates. In the present study, these variables were unaccounted for. It is also conceivable that with a larger sample size, the effects of the menstrual phases on sleep would become more obvious. Hence, power limitations should be taken into consideration when interpreting the present results.

## Conclusions

The current study in junior endurance athletes is the first to examine sex differences in sleep and sleep stage distributions with objective sleep measures, and provides the longest overview of objective sleep parameters to date. We showed that female athletes sleep better (longer TST, LS (h) and REM (h, %), shorter REML and higher SE) than males on several objective sleep parameters. However, female athletes tended to show poorer subjective sleep quality, which could not be ascribed to sex differences in mental and physical loads. Exploring the role of the menstrual cycle and its phases on female athletes' sleep over two consecutive months, we showed that menstrual bleeding days were associated with increases in TIB and SWS (h), and decreases in LS (%), compared to non-bleeding days; that menstrual phase was associated with decreased LS (h), SWS (%) and REML compared to pre-menstrual phase; and that follicular phase was associated with decreased SE and increased SWS (%) compared to luteal phase. Taken together, these findings indicate that the menstrual cycle may induce an additional load in the female athletes, and thus cause variations in the need for physiological recovery. This is likely due to the effect of the different hormones on physiological systems, discomfort and blood loss associated with menstrual bleeding, a need for more "downtime" during menstrual bleeding, or due to other menstrual symptoms that cause sleep fragmentation or disruption. Overall, this study extends existing research by providing long-term data of sex differences in sleep stage distributions and as such provides novel insight into the role of the menstrual cycle on sleep among female athletes.

## Supporting information

**S1 Fig. Division of the menstrual cycle into phases.** An illustrative example of the division of the menstrual cycle into menstrual, pre-menstrual, follicular and luteal phases investigated in this study.
(TIF)

**S1 Table. Calculation of training load.** The intensity scale used in this study to determine the training load of endurance and strength, plyometric and speed training.
(DOCX)

**S2 Table. The well-being questionnaire.** The questionnaire included 5 items, given to the participants in Norwegian, scored on a visual analogue Likert scale ranging from 0 (low well-being) to 10 (high well-being).
(DOCX)

**S1 File. Sex differences in sleep data.**
(SAV)

## Acknowledgments

Athletes' participation is deeply appreciated. We thank research assistants Emilie F.W. Raanes and Maja Olsen for help with data collection, Ignacio Polti for help with figures, and Dr. Dionne Noordhof for valuable discussions about the methodology used in this study.

## Author Contributions

**Conceptualization:** Maria Hrozanova, Christian A. Klöckner, Øyvind Sandbakk, Ståle Pallesen, Frode Moen.

**Data curation:** Maria Hrozanova, Christian A. Klöckner.

**Formal analysis:** Maria Hrozanova, Christian A. Klöckner.

**Investigation:** Maria Hrozanova, Frode Moen.

**Methodology:** Maria Hrozanova, Christian A. Klöckner, Øyvind Sandbakk, Ståle Pallesen, Frode Moen.

**Project administration:** Øyvind Sandbakk, Frode Moen.

**Resources:** Christian A. Klöckner, Øyvind Sandbakk, Frode Moen.

**Supervision:** Christian A. Klöckner, Øyvind Sandbakk, Ståle Pallesen, Frode Moen.

**Visualization:** Maria Hrozanova.

**Writing – original draft:** Maria Hrozanova.

**Writing – review & editing:** Maria Hrozanova, Christian A. Klöckner, Øyvind Sandbakk, Ståle Pallesen, Frode Moen.

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
