## [Decision Letter · Decision Letter 0]

22 Mar 2021

PONE-D-20-31663

Sex differences in sleep and influence of the menstrual cycle on women’s sleep in junior endurance athletes

PLOS ONE

Dear Dr. Hrozanova,

Thank you for submitting your manuscript to PLOS ONE. After careful consideration, we feel that it has merit but does not fully meet PLOS ONE’s publication criteria as it currently stands. Therefore, we invite you to submit a revised version of the manuscript that addresses the points raised during the review process.

The two peer reviewers have included detailed comments to consider in your revision. 

We look forward to receiving your revised manuscript.

Kind regards,

Melissa M Markofski

Academic Editor

PLOS ONE

Journal Requirements:

2. Please provide additional details regarding participant consent.

In the ethics statement in the Methods and online submission information, please ensure that you have specified what type you obtained (for instance, written or verbal, and if verbal, how it was documented and witnessed). If your study included minors, state whether you obtained consent from parents or guardians. If the need for consent for participants under 18 was waived by the ethics committee, please include this information.

4. In your Methods section, please provide additional information about the participant recruitment method and the demographic details of your participants. Please ensure you have provided sufficient details to replicate the analyses such as:

- the recruitment date range (month and year)

- a description of any inclusion/exclusion criteria that were applied to participant recruitment

- a table of relevant demographic details

- a statement as to whether your sample can be considered representative of a larger population

- a description of how participants were recruited

- descriptions of where participants were recruited and where the research took place.

'..The authors acknowledge the financial support of the Centre for Elite Sports Research.'

 'The author received no specific funding for this work.'

7. Please ensure that you refer to Figure 2 in your text as, if accepted, production will need this reference to link the reader to the figure.

Reviewers' comments:

Reviewer's Responses to Questions

**Comments to the Author**

1. Is the manuscript technically sound, and do the data support the conclusions?

Reviewer #1: Yes

Reviewer #2: No

2. Has the statistical analysis been performed appropriately and rigorously? 

Reviewer #1: Yes

Reviewer #2: No

3. Have the authors made all data underlying the findings in their manuscript fully available?

Reviewer #1: Yes

Reviewer #2: Yes

4. Is the manuscript presented in an intelligible fashion and written in standard English?

Reviewer #1: Yes

Reviewer #2: Yes

5. Review Comments to the Author

Reviewer #1: The authors measured sleep through radar in male and female youth athletes across 61 days. They report comparisons between male and female, and between various menstrual-cycle related time-points. Their conclusion is that females do have better objective but worse subjective sleep than males, that women spent longer time in bed and in deeper sleep during menses vs. non-menses, lower sleep efficiency during the follicular vs. luteal phase and lower light sleep in menstrual vs. pre-menstrual phase. The authors conclude recovery may be increased during menstrual and pre-menstrual times.

The manuscript is well-organised, well-written, correctly interpreted and adequately addresses limitations. The results are important, novel and applicable as well as paving the way for future research.

I have few and very minor suggestions:

1) Line 237, should be Figure 2

2) Line 307, I assume this is just supposition and not evidence-based? If the latter, please add reference.

The major comment, which the authors have addressed, is that they did not confirm ovulatory cycles. This does not invalidate results, but does mean they need replicating (with confirmation) and/or taking with precaution.

Reviewer #2: This manuscript describes sex difference in sleep over 61 days in a small sample of young Norwegian athletes that included 15 girls who also tracked their menstrual cycles. The introduction sets up the research aims, and the purpose is clear. Methods included radar-sleep tracking (Somnofy sleep monitor developed and validated in Norway) for objective sleep duration and sleep stages. The Pittsburgh Sleep Quality Index (PSQI) was used for self-report sleep quality over the past month. Measures are well described for menstrual cycle phases by diary reports of menses days. Analyses included multivariate modeling. Results support past research that women score worse on subjective PSQI sleep quality but better on objective sleep measures. Girls had longer sleep duration than boys, and sex accounted for 25% of the variance (R-square = .25) in Total Sleep Time (TST), and 12.6% of the variance (R-square = .126) in Sleep Efficiency (SE). Sex explained 6.1% of the variance in PSQI sleep quality scores (R-square = .061). For the 15 female athletes, menstrual cycle changes in objective sleep were either less evident, with low variance explained, or sample was indeed too small to obtain statistical significance; they were in bed longer and had less sleep efficiency during the follicular phase; these two variables have obvious collinearity. Results are discussed in light of prior studies, in particular one study by Driver et al. (#21). Limitations are adequately addressed. I do have some specific comments for authors to consider:

1. Sleep stages defined in Table 1 are for minutes. Given the sex differences in TST, one would expect a longer sleep duration to include more minutes in each sleep stage. Standardizing each night’s time spent in sleep stages as a percentage of their TST would better support the interpretation of the results about sleep stages.

2. The Somnofy system was not particularly valid for sleep onset latency in a prior publication, but please indicate the criterion for minutes to onset – it could be lights-out to first stage of LS, first stage of SWS, or first 1, 2, 10 minutes of LS, for example. Do authors have REM onset latency as well?

3. In light of the discussion, please indicate if women took pain medication during menses.

4. Table 1 would be more useful to readers if it was specific to hours or minutes (rather than “time”). Because SE refers to sleep efficiency, the S.E. columns in table 2 and table 3 should be spelled out or distinguished in some way. This would be a useful place to add criterion for sleep onset latency, and to add REM onset latency.

4. There are 2 tables labeled as Table 2. The first table 2 is of less interest and could be supplemental. The second Table 2 with R-square values in the last column needs to be more clear – please provide the total R-square at the end, if all of these were in the same model, but it is unclear if they are separate models because of the collinearity between all these sleep variables. The R-square percentages in Table 2 or Table 3 are confusing, unless the heading in the column is meant to say "percent variance explained" rather than the actual "R-square" value. Italicizing significant p-values is very difficult to see in tables; with p-values provided, italics are not needed in that column. An asterisk or bolding would be better, depending on journal requirements.

5. ICC values need a little more explanation and discussion. They are very low in Tables 2 and 3, and correlations above .80 would be expected for validity and reliability in night-to-night sleep parameter consistency. Please indicate how many nights are represented with these values and why they are important to include in results for the regression model(s).

6. Figure 1 is not particularly useful and could be supplemental material.

7. Figure 2 is very informative showing differences between men and women. It would be helpful to include the women’s data by menstrual cycle phase as part of this figure or as a new figure.

8. Please include Cronbach alpha for PSQI in your sample and whether it differed by sex. The absence of statistically significant differences may reflect the small N for women (n=15) and perhaps effect sizes should be included.

9. Discussion may require a little revision if results about sleep stage percentages differ from minutes. Discussion mentions levels of discomfort, which raises the issue of medications taken by the women during menses.

10. Discussion of comparisons with the prior research is included with possible explanations for differences, but comparisons focus on Driver #21 findings only. Thus, it is not clear if this study supports other studies that also conflict with Driver’s findings, or if Driver’s results also conflict with others. Driver’s participants were not in their own home, and had more uncomfortable PSG electrodes every-other night in a laboratory, creating the potential for first-night effects, so comparisons of SE would be useful in the discussion; was their overall SE worse or better than women in this sample studied at home with less invasive equipment? It would help readers if discussion compared results with other studies besides Driver’s one study, for example, Baker et al PSG in the lab, Lee et al PSG in the lab (1990) or PSG in the home (2000), or Parry et al PSG young women controls.

11. The conclusion that this manuscript provides “long-term” data is a bit of a stretch, as that would usually mean over more time than just 2 menstrual cycles, but enough to have, for example, PSQI self-report at 6-month intervals.

6. PLOS authors have the option to publish the peer review history of their article (what does this mean?). If published, this will include your full peer review and any attached files.

Reviewer #1: **Yes: **Toby Mundel

Reviewer #2: No

---

## [Author Response · Author response to Decision Letter 0]

22 Apr 2021

We thank the Academic Editor for considering our manuscript, and to the Reviewers for conducting a thorough review. We have responded to all your comments, and believe that your feedback improved the quality of this manuscript. Please find our responses to each comment in the .docx file named "Response to Reviewers".

---

## [Decision Letter · Decision Letter 1]

4 Jun 2021

Sex differences in sleep and influence of the menstrual cycle on women’s sleep in junior endurance athletes

PONE-D-20-31663R1

Dear Dr. Hrozanova,

We’re pleased to inform you that your manuscript has been judged scientifically suitable for publication and will be formally accepted for publication once it meets all outstanding technical requirements. Reviewer 2 had one helpful minor comment that you may wish to consider for your final manuscript submission. 

Kind regards,

Melissa M Markofski

Academic Editor

PLOS ONE

Additional Editor Comments (optional):

Reviewers' comments:

Reviewer's Responses to Questions

**Comments to the Author**

1. If the authors have adequately addressed your comments raised in a previous round of review and you feel that this manuscript is now acceptable for publication, you may indicate that here to bypass the “Comments to the Author” section, enter your conflict of interest statement in the “Confidential to Editor” section, and submit your "Accept" recommendation.

Reviewer #1: All comments have been addressed

Reviewer #2: All comments have been addressed

2. Is the manuscript technically sound, and do the data support the conclusions?

Reviewer #1: (No Response)

Reviewer #2: Yes

3. Has the statistical analysis been performed appropriately and rigorously? 

Reviewer #1: (No Response)

Reviewer #2: Yes

4. Have the authors made all data underlying the findings in their manuscript fully available?

Reviewer #1: (No Response)

Reviewer #2: Yes

5. Is the manuscript presented in an intelligible fashion and written in standard English?

Reviewer #1: (No Response)

Reviewer #2: Yes

6. Review Comments to the Author

Reviewer #1: (No Response)

Reviewer #2: This revised manuscript has improved in clarity with the better explanation of ICC for readers, and with the percentage of sleep stages analyzed. The discussion has more depth, and limitations are adequately addressed. The tables and figures are more logical, and adding units as a column to Table 1 will be more helpful to readers. I do have a minor comment: On Line 468, rather than “divergences” I would suggest “variations”. This was an ambitious and rigorous research endeavor and the manuscript would make a significant contribution to knowledge about young women and sleep.

7. PLOS authors have the option to publish the peer review history of their article (what does this mean?). If published, this will include your full peer review and any attached files.

Reviewer #1: No

Reviewer #2: No

---

## [Editor Report · Acceptance letter]

8 Jun 2021

PONE-D-20-31663R1 

Sex differences in sleep and influence of the menstrual cycle on women’s sleep in junior endurance athletes 

Dear Dr. Hrozanova:

I'm pleased to inform you that your manuscript has been deemed suitable for publication in PLOS ONE. Congratulations! Your manuscript is now with our production department. 

Kind regards, 

on behalf of

Dr. Melissa M Markofski 

Academic Editor

PLOS ONE